# Hydrochemistry Differences and Causes of Tectonic Lakes and Glacial Lakes in Tibetan Plateau

**Meiping Sun [1,2], Huian Jin [1,3,*], Xiaojun Yao [1], Luxia Yan [1], Xiaofeng Li [1] and Yongpeng Gao [4]**

[1]  College of Geography and Environment Science, Northwest Normal University, Lanzhou 730070, China; sunmeiping1982@nwnu.edu.cn (M.S.); xj_yao@nwnu.edu.cn (X.Y.); 18893848965@163.com (L.Y.); lixiaofeng_geo@163.com (X.L.)

[2]  State Key Laboratory of Cryosphere Science, Northwest Institute of Eco-Environment and Resources, Chinese Academy of Sciences, Lanzhou 730000, China

[3]  Gansu Forestry Polytechnic, Tianshui 741020, China

[4]  Institute of International Rivers and Eco-Security, Yunnan University, Kunming 650091, China; 22019000220@mail.ynu.edu.cn

*  Correspondence: 2018222406@nwnu.edu.cn; Tel.: +86-0931-797-1161

**Abstract:** The Tibetan Plateau has the largest lake cluster in China and in the world. In order to clarify the differences of lake hydrochemistry of Tibetan Plateau, water samples were collected from 32 lakes, including 22 tectonic lakes and 11 glacial lakes, along the Tibetan Plateau road, from September to October 2016. We detected and analyzed the major ion concentrations and characteristics of samples, and discuss the hydrochemistry type, controlling factors, and major ion sources of lake water. The results showed that, firstly, tectonic lake samples on the Tibetan Plateau have much higher physicochemical parameters and ion contents than glacial lakes, and Total Dissolved Solids (TDS) contents fluctuate from high to low latitudes. The variations of ion concentrations in the northern part of the Qiagui Co were more fluctuating and have two obvious peaks, while the variations in the southern part were moderate. The TDS of glacial lakes were low and leveling off in the upper and middle reaches of the basin, while higher and more variable in the lower reaches. Secondly, the tectonic lakes were mainly chloride saline lakes, with $Na^+$ as the major cation, and $SO_4^{2-}$, $Cl^-$ as the major anions. Glacial lakes were mainly carbonate and sulfate type lakes, $Ca^{2+}$ and $Mg^{2+}$ were the major cations, $HCO_3^-$ was the major anion, and $SO_4^{2-}$ was the second. Thirdly, the hydrochemistry processes of the tectonic lakes were mainly controlled by evaporation-crystallization, and the ions mainly came from the evaporites of basin. Glacial lake water samples were mainly influenced by the weathering of basin rocks, with ion sources strongly influenced by the weathering of basin carbonates than evaporites, with calcite and dolomite being important sources of $Ca^{2+}$, $Mg^{2+}$, and $HCO_3^-$.

**Keywords:** hydrochemistry process; major ion characteristics; tectonic lakes; glacial lakes; Tibetan Plateau

## 1. Introduction

Ion characteristics is an important aspect of hydrochemistry research, and its composition is influenced by many factors, such as the natural processes, climatic conditions, and ecological environment of the areas it passes through. Evaporation–crystallization of confined water bodies, regional rock weathering, and atmospheric precipitation constitute the three main factors controlling the hydrochemical of surface water bodies [1]. The characteristics and distribution patterns of hydrochemistry reflected the influence of atmospheric deposition, basin lithology, vegetation cover, human activities, crustal movements, and other factors on basins, and record important information related to the water cycle, such as the source, transport, and transformation of solutes in the water [2].

As an important part of surface water bodies, lakes are involved in the natural hydrological cycle and are important information carriers for the study of climate change, environmental change, and the water cycle [3–6].

The Tibetan Plateau is known as the "The third Pole of the Earth", and has the highest, largest, and most densely distributed alpine lake cluster in the world [7,8]. As an important water body of the Tibetan Plateau, the study on the hydrochemical characteristics of lakes has attracted much attention, and a lot of research has been carried out [9–12]. For example, Guo [13] carried out continuous point sampling of the near-shore surface waters of Nam Co Lake in 2006–2010. The results showed that most of the ion concentrations in the lake water affected by evaporation, precipitation, and runoff, showed obvious seasonal variations, with higher concentrations during the monsoon period (June–September) and lower concentrations during the non-monsoon period (January–April), especially during the freeze period. Wang [14] explored the ion sources, host factors, and hydrochemistry evolution trends of lake water in Daggyaima Co, which revealed the evolution of lake water from freshwater to saltwater on the Tibetan Plateau. Li [15] analyzed the chemical composition, ion sources, and host factors in water samples of lakes (Pung Co, Ngamring Co, Daggyai Co, Daggyaima Co) and tributaries in some small basins on the Tibetan Plateau, and pointed out that there were differences in hydrochemistry and material sources of different lakes and their tributaries. However, the Tibetan Plateau lake hydrochemistry researches are mostly based on a certain lake or basin at present, and concentrate on the Qinghai Lake [16], Nam Co [13], Yamzho Yumco [17], Mapam Yumco [18], and some other big lakes, but few comparative studies involve different types of lakes.

According to the lake origin classification, the Tibetan Plateau distribution is more tectonic lakes and glacial lakes. Tectonic lakes are various tectonic depressions formed by the internal forces of crustal movements, including crustal depressions, depressions, and subsidence produced by geological tectonic movements. Tectonic lakes are all kinds of tectonic depressions formed by the movement of the earth's crust, through water storage and the formation of lakes, generally for large and medium-sized closed lakes, steep shores, and along the tectonic line development, the lake is very deep. Glacial lakes are a class of lakes made up of glacial troughs blocked by pits and moraines eroded by glaciers. It is mainly distributed in high mountain glaciers, such as Nyainqêntanglha Mountains and Himalaya Mountains, with higher altitudes, smaller lakes, more outlets, and an area of less than 10 km$^2$ [19]. In this study, 21 tectonic lakes on the Tibetan Plateau and 11 glacial lakes in the Yairu Zangbo basin of Himalaya Mountains, were studied. The major ion characteristics of lake waters were analyzed, and the differences in hydrochemistry between tectonic lakes and glacial lakes on the Tibetan Plateau and their controlling factors were revealed, providing a scientific reference for the study of surface hydrochemistry processes on the Tibetan Plateau.

## 2. Study Area

The Tibetan Plateau is located in China (26°00′ N~39°46′ N, 73°18′ E~104°46′ E), with an area of 257.24 × 10$^4$ km$^2$, stretching from the Pamir Plateau in the west, to the Hengduan Mountains in the east, the southern edge of the Himalayas Mountains in the south, and the Kunlun Mountains and the Qilian Mountains in the north [20]. The Tibetan Plateau is administratively divided into six provinces (autonomous regions): Xinjiang, Tibet, Qinghai, Yunnan, Sichuan, and Gansu, with the main part in Qinghai and Tibet [21]. Due to the differences in geological features and development history, the territory of the Tibet Autonomous Region includes the Sivarik, Himalayan, Gangdisê-Nyainqêntanglha, Qiangtang-Qamdo, and South Kunlun-Bayan Har five stratigraphic regions [22], and four stratigraphic regions of Qilian, Kunlun, Bayan Har-Qinling, and Tanggula in Qinghai Province [23].

The 32 lakes studied belong to four stratigraphic regions, shown in Table 1: (a) Gangdisê-Nyainqêntanglha stratigraphic region, it can be further divided into two stratigraphic zones of Coqên-Xainza and Baingoin, the former is mainly shale and limestone interbedded, the upper part is dolomitic limestone, and the lower part is mainly shale; the latter is a set of shallow

sea-coastal phase carbonate and clastic rocks, the upper part is quartzite with a little sandy slate and siliceous slate, the middle part is quartzite with carbonate rocks, and lower part is carbonate rocks. (b) Qiangtang-Qamdo stratigraphic region, its lithology mainly limestone, biotite limestone, intercalated quartz sandstone and siltstone. (c) Qilian stratigraphic region, where the ancient basement formed during the Ediacaran period was concentrated, with strong metamorphic effect, on which the Ediacaran were not integrated by angle, and were shallow metamorphic clastic rocks, carbonate rocks, and volcanic eruption deposits. (d) Himalaya stratigraphic region, where metamorphic rocks, such as potassium feldspar, oblique feldspar, marble, quartz sandstone, siltstone, shale, and limestone are commonly developed [24]. We obtained samples from 32 lakes belonging to four stratigraphic regions along the route based on the planned route and the accessibility of the lake locations. Table 1 also lists the location of the lake centroid and other parameters. The lake parameters, such as recharge coefficient, mean annual precipitation, and mean annual temperature of the lake, were taken from [25].

**Table 1.** The parameters of tectonic lakes and glacial lakes in Tibetan Plateau.

| Type | No. | Name | Str | Lat | Lon | Alt | Area | Rec | Pre | Tem |
|------|-----|------|-----|-----|-----|-----|------|-----|-----|-----|
| | 1 | Pung Co | | 31.51 | 90.97 | 4539 | 176.5 | 7.5 | 300~400 | −2 |
| | 2 | Bangkog Co | | 31.74 | 89.51 | 4533 | 123.9 | 38.4 | 300 | −2~−1 |
| | 3 | Yaggain Co | | 31.56 | 89.01 | 4546 | 112.4 | 15.9 | 150~200 | −2~0 |
| | 4 | Co Ngoin | | 31.59 | 88.72 | 4570 | 268.4 | 22.6 | 200~300 | −2~0 |
| | 5 | Siling Co | a | 31.81 | 88.99 | 4552 | 2300.3 | 24.9 | 300 | 0 |
| | 6 | Serbug Co | | 32.01 | 88.22 | 4531 | 92.9 | 32.9 | 200~300 | −2~0 |
| | 7 | Qiagui Co | | 31.82 | 88.25 | 4560 | 88.9 | 1668.1 | / | −2 |
| | 8 | Dagzê Co | | 31.89 | 87.52 | 4478 | 311.0 | 44.5 | 200 | 0~2 |
| | 9 | Dong Co | | 32.18 | 84.74 | 4402 | 105.4 | 62.1 | 150~200 | 0 |
| Tectonic lake | 10 | Mubu Co | | 33.95 | 85.34 | 4822 | 14.0 | / | / | / |
| | 11 | Daxiong Lake | b | 34.05 | 85.61 | 4893 | 42.9 | / | / | / |
| | 12 | Xueyuan Co | | 34.25 | 85.72 | 5213 | 24.8 | 7.5 | 100~150 | −6~−4 |
| | 13 | Qido Co | | 31.23 | 85.08 | 4635 | 9.4 | / | / | / |
| | 14 | Dawa Co | | 31.24 | 84.96 | 4632 | 118.2 | 21.2 | 200~300 | 0 |
| | 15 | Daggyai Co | a | 29.84 | 85.72 | 5152 | 109.4 | 5.5 | 300 | 2 |
| | 16 | Daggyaima Co | | 29.65 | 85.74 | 5069 | 10.0 | / | / | / |
| | 17 | Ngamring Co | | 29.31 | 87.19 | 4304 | 21.1 | 7.0 | 300~400 | 4~6 |
| | 18 | Lang Co | | 29.21 | 87.39 | 4299 | 12.1 | 7.0 | 300~400 | 6 |
| | 19 | Xiao Qadam | | 37.49 | 95.51 | 3178 | 88.1 | 86.1 | 80 | 1 |
| | 20 | Gahai Lake | c | 37.13 | 97.55 | 2853 | 34.8 | 61.0 | 100 | 2~4 |
| | 21 | Qinghai Lake | | 36.89 | 100.20 | 3197 | 4348.1 | 5.8 | 330 | 1 |
| | 22 | A | | 28.22 | 87.66 | 5648 | | | | |
| | 23 | B | | 28.19 | 87.64 | 5366 | | | | |
| | 24 | C | | 28.24 | 87.66 | 5599 | | | | |
| | 25 | D | | 28.17 | 87.62 | 5383 | | | | |
| | 26 | E | | 28.33 | 87.83 | 4205 | | | | |
| Glacial lake | 27 | F | d | 28.33 | 87.78 | 4199 | | | | |
| | 28 | G | | 28.32 | 87.81 | 4194 | | | | |
| | 29 | H | | 28.15 | 87.96 | 4492 | | | | |
| | 30 | I | | 28.18 | 87.86 | 4302 | | | | |
| | 31 | J | | 28.18 | 87.79 | 4272 | | | | |
| | 32 | K | | 28.28 | 87.81 | 4221 | | | | |

Str indicates the stratigraphic region in which the lake is located. Lat and Lon are latitude and longitude in decimal degrees, respectively. Alt stands for water surface elevation of the lake in meters. The unit of lakes' area are $km^2$. Rec is the lake recharge coefficient. Cat represents the catchment area in $km^2$. Pre represents mean annual precipitation. Tem is mean annual temperature in the lake district, in °C. "/" indicates that the parameter was not retrieved.

## 3. Materials and Method

### 3.1. Sampling and Measurement

A field study was done of 21 tectonic lakes along national highway G315 in the northeast, provincial road S301, S302, and S206 in the center, and national highway G219, G318 in the south of the Tibetan Plateau, and 11 glacial lakes in the Yairu Zangbo basin on the northern slopes of the Himalayas in September–October 2016. Two to six water samples were collected from each lake using polyethylene plastic bottles at different locations along the lake shore, according to the lake area.

Two water samples were collected from lakes with an area less than 100 km$^2$, 4 samples were collected from lakes with an area between 100 km$^2$ and 500 km$^2$, 6 samples were collected from lakes with an area larger than 500 km$^2$, and, finally, we averaged the results of multiple water sample detections from each lake. Lake water sampling location distribution is shown in Figure 1. Lake water samples were collected by first rinsing polyethylene plastic bottles with lake water three times before being loaded, sealed, and refrigerated immediately in a portable refrigerator. All samples were shipped back to the Eco-Hydrological Processes Laboratory of Northwestern Normal University by freezing, and were stored in a −15 °C cryogenic chamber, where they were naturally thawed at room temperature (about 25 °C) for 48 h before testing.

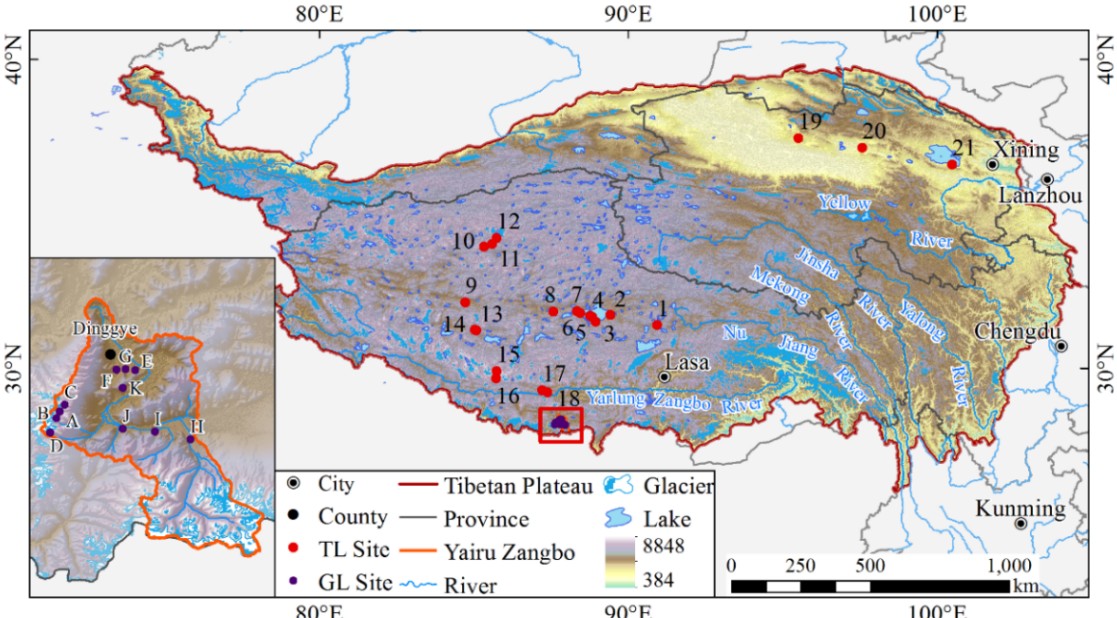

**Figure 1.** Location distribution of the 32 lakes in this study. Tectonic lake (TL) Sites in the figure are the locations of the tectonic lake sample sites, same no. as in Table 1. Glacial lake (GL) Sites in the figure are the locations of the glacial lake sample sites.

The pH was detected using a pH meter (pHSJ-4A, Shanghai Yidian Scientific Instrument Co., Ltd., Shanghai, China), detection range 0.000~14.000, with an accuracy of ±0.05%. Electrical Conductance (EC) was detected using a conductivity meter (DDSJ-308A, Shanghai Yidian Scientific Instrument Co., Ltd., Shanghai, China) at a constant sample temperature of 25 °C, detection range 0.00 μS/cm~199.9 mS/cm, with an accuracy of ±0.5%. Prior to ion concentration determination, all samples were filtered with 0.45 μm filter membranes. Major cations ($Na^+$, $K^+$, $Mg^{2+}$, $Ca^{2+}$) were detected with an ion chromatograph (Dionex-600), and major anions ($Cl^-$, $NO_3^-$, $SO_4^{2-}$) were detected with Dionex-300 ion chromatograph; the accuracy of ion determination was ng/g level, and the testing error was less than 5%. Due to the large amounts of samples required for the detection of $HCO_3^-$, there is no such ion concentration in the detection index, so this study used the charge balance method to calculate the $HCO_3^-$ ion concentration [26,27]. We compared dissolved cation equivalents ($TZ^+ = 2Ca^{2+} + 2Mg^{2+} + Na^+ + K^+$) to the detected anion equivalents ($TZ^- = Cl^- + NO_3^- + 2SO_4^{2-}$), and calculated the $HCO_3^-$ ion concentration. Total Hardness (TH) indicates the content of calcium and magnesium ions in the water column, which can be calculated by the following formula [28]: TH (g/L) = 2.497 × Ca (g/L) + 4.118 × Mg (g/L). Total Dissolved Solids (TDS) content was calculated as the sum of the ion concentrations [29].

### 3.2. Piper Diagram

The Piper diagram is a method for classifying water samples; it was first proposed by Piper in 1953 [30]. The chemical type of the surface water is classified by calculating the concentration ratios of the major anions and cations in the surface water and determining the position of the sample in the Piper diagram. The graph, which can be plotted using Origin 2018 software, consists of a rhombus and two equilateral triangles that discuss the proportion of major ions in the anion and cation separately, while the rhombus relates the anions and cations, thus, reflecting the compositional characteristics of the ions and the type of water chemistry in the surface water.

### 3.3. Ion Correlation Analysis

The correlation analysis of ions in water can effectively reveal the correlation of hydrochemical components, and infer whether the ions have the same origin. The SPSS 23 software was used in the analysis to obtain the Pearson correlation coefficients (at 0.01 and 0.05 levels) by bivariate analysis.

### 3.4. Gibbs Diagram

Gibbs diagrams provide a clear picture of the water chemistry of surface water and provide a basis for studying the chemical composition of surface water and the causes of formation; thus, determining the main sources of ions in the surface water. The diagram was plotted using Origin 2018 software; the vertical coordinates of the figure show the TDS content in logarithmic form, and the horizontal coordinates show the ion concentration ratios in terms of $Na^+/(Na^+ + Ca^{2+})$ or $Cl^-/(Cl^- + HCO_3^-)$.

### 3.5. Ion Ratio Analysis

Ion combination and ratio method for the analysis of sources of weathering products in river water, proposed by Gaillardet in 1999, was based on a study of 60 large rivers around the world [31]. Statistical analysis revealed that there are three main sources of weathering products in river water: carbonate products, silicate products, and evaporation salts. The study analyzed the contribution of various rocks to ionic composition by anion and cation binding and ratios, and used Origin 2018 Software for mapping.

## 4. Results and Discussion

### 4.1. Ion Characteristics of Tectonic Lakes

The TDS of the tectonic lakes in this study ranged 0.14~79.34 g/L with mean value of 17.32 g/L. The pH ranged 7.83~10.12, and mean value of 9.09. EC variation ranged 0.18~96.10 mS/cm at 25 °C, with a mean of 19.32 mS/cm. TH ranges were 0.08~11.36 g/L in soft water to very hard water (Table 2).

The tectonic lake samples are shown in the Piper diagram (Figure 2a); the lake samples were mostly close to $Na^+ + K^+$, and mostly in the high value region, except for the Daggyaima Co, indicating that $Na^+ + K^+$ were absolutely dominant in tectonic lake water samples, and $Ca^{2+}$, $Mg^{2+}$ content is low. Among the anions, some of the samples were at the high value region of $HCO_3^-$, while the rest were near the $SO_4^{2-}$ (about 10%~60%) and $Cl^-$ (about 20%~80%) terminal.

Hydrochemistry types can be classified into four types based on the Piper diagram: $Ca(Mg)-SO_4$ (region A), $Ca-HCO_3$ (region B), $Na(K)-Cl$ (region C), and $Na-HCO_3$ (region D) [32]. The evolution of the Tibetan Plateau lakes, generally experienced from freshwater lakes to saline lakes, and then to salt lakes, until the end of the dry salt lake evolution, along with its hydrochemistry type, also changed, with a range from carbonate-sulfate-chloride types. Among the 21 tectonic lakes in this study, 12 were $Na(K)-Cl$ type, 5 were $Na-HCO_3$ type, and 4 were $Ca-HCO_3$ type, indicating that the tectonic lakes on Tibetan Plateau are mostly chloride-type lakes and are in the later stage of lake evolution.

Table 2. The physicochemical indexes and major ion concentration of tectonic lakes and glacial lakes in Tibetan Plateau.

| Type | Item | pH | EC | $Cl^-$ | $NO_3^-$ | $SO_4^{2-}$ | $HCO_3^-$ | $Na^+$ | $K^+$ | $Mg^{2+}$ | $Ca^{2+}$ | TH | TDS |
|------|------|-----|-----|------|------|------|------|------|------|------|------|------|------|
| | 1 | 9.90 | 15.27 | 0.72 | - | 3.66 | 7.05 | 4.03 | 0.37 | 0.30 | 0.04 | 1.36 | 16.18 |
| | 2 | 9.15 | 0.48 | 0.01 | - | 0.01 | 0.35 | 0.04 | 0.01 | 0.04 | 0.02 | 0.21 | 0.48 |
| | 3 | 9.44 | 14.88 | 2.07 | - | 4.24 | 2.42 | 3.37 | 0.36 | 0.34 | 0.04 | 1.50 | 12.84 |
| | 4 | 8.08 | 0.43 | 0.01 | - | 0.02 | 0.28 | 0.03 | - | 0.04 | 0.02 | 0.20 | 0.40 |
| | 5 | 9.41 | 13.86 | 1.90 | - | 3.90 | 1.95 | 3.03 | 0.32 | 0.30 | 0.03 | 1.32 | 11.44 |
| | 6 | 9.19 | 3.94 | 0.44 | - | 0.90 | 0.76 | 0.74 | 0.08 | 0.10 | 0.02 | 0.47 | 3.04 |
| | 7 | 8.30 | 0.44 | 0.01 | - | 0.03 | 0.27 | 0.04 | 0.01 | 0.02 | 0.03 | 0.17 | 0.42 |
| | 8 | 10.12 | 22.90 | 1.03 | - | 4.53 | 12.84 | 6.06 | 0.53 | 0.57 | 0.18 | 2.81 | 25.75 |
| | 9 | 9.06 | 65.80 | 11.04 | - | 26.47 | 5.30 | 15.45 | 3.15 | 2.29 | 0.12 | 9.73 | 63.83 |
| | 10 | 8.77 | 91.30 | 31.56 | - | 7.46 | 5.52 | 21.12 | 2.54 | 1.67 | 0.28 | 7.58 | 70.14 |
| | 11 | 9.19 | 21.80 | 4.31 | - | 4.24 | 3.52 | 4.61 | 0.24 | 0.71 | 0.04 | 3.02 | 17.68 |
| | 12 | 9.00 | 3.40 | 0.34 | - | 0.74 | 1.00 | 0.54 | 0.04 | 0.19 | 0.03 | 0.85 | 2.87 |
| | 13 | 9.32 | 2.72 | 0.16 | - | 0.71 | 0.95 | 0.39 | 0.06 | 0.19 | 0.01 | 0.81 | 2.47 |
| Tectonic | 14 | 9.63 | 16.19 | 1.13 | 0.01 | 6.81 | 3.95 | 4.01 | 0.36 | 0.64 | 0.03 | 2.72 | 16.93 |
| lake | 15 | 9.33 | 4.50 | 0.21 | - | 0.82 | 1.93 | 1.00 | 0.06 | 0.11 | 0.03 | 0.54 | 4.17 |
| | 16 | 7.83 | 0.18 | - | - | 0.04 | 0.06 | - | - | - | 0.03 | 0.08 | 0.14 |
| | 17 | 9.85 | 8.55 | 0.36 | - | 1.42 | 9.03 | 3.97 | 0.02 | 0.07 | 0.20 | 0.78 | 15.06 |
| | 18 | 9.46 | 2.77 | 0.13 | - | 0.02 | 1.98 | 0.60 | 0.01 | 0.11 | 0.01 | 0.49 | 2.87 |
| | 19 | 8.36 | 96.10 | 33.52 | - | 13.85 | 4.10 | 24.39 | 0.43 | 2.30 | 0.75 | 11.36 | 79.34 |
| | 20 | 8.27 | 1.91 | 0.35 | 0.01 | 0.22 | 0.35 | 0.23 | 0.01 | 0.05 | 0.13 | 0.51 | 1.33 |
| | 21 | 9.13 | 18.33 | 5.14 | - | 1.80 | 3.03 | 3.39 | 0.14 | 0.94 | 0.06 | 4.01 | 14.50 |
| | Min | 7.83 | 0.18 | - | - | 0.01 | 0.06 | - | - | - | 0.01 | 0.08 | 0.14 |
| | Max | 10.12 | 96.10 | 33.52 | - | 26.47 | 12.84 | 24.39 | 3.15 | 2.30 | 0.75 | 11.36 | 79.34 |
| | Mean | 9.09 | 19.32 | 4.50 | - | 3.90 | 3.17 | 4.62 | 0.42 | 0.52 | 0.10 | 2.41 | 17.23 |
| | Std | 0.60 | 27.98 | 9.62 | - | 6.05 | 3.22 | 6.89 | 0.83 | 0.70 | 0.16 | 3.16 | 23.26 |
| | Median | 9.19 | 8.55 | 0.44 | - | 1.42 | 1.98 | 3.03 | 0.08 | 0.19 | 0.03 | 0.85 | 11.44 |
| | 25% quartile | 8.57 | 2.32 | 0.15 | - | 0.13 | 0.56 | 0.31 | 0.01 | 0.06 | 0.02 | 0.48 | 1.90 |
| | 75% quartile | 9.45 | 20.07 | 3.19 | - | 4.39 | 4.70 | 4.32 | 0.37 | 0.68 | 0.12 | 2.92 | 17.30 |

**Table 2.** *Cont.*

| Type | Item | pH | EC | Cl$^-$ | NO$_3^-$ | SO$_4^{2-}$ | HCO$_3^-$ | Na$^+$ | K$^+$ | Mg$^{2+}$ | Ca$^{2+}$ | TH | TDS |
|---|---|---|---|---|---|---|---|---|---|---|---|---|---|
| | A | 7.73 | 0.08 | - | - | 0.02 | 0.02 | - | - | - | 0.01 | 0.03 | 0.05 |
| | B | 8.53 | 0.02 | - | - | - | 0.01 | - | - | - | - | 0.01 | 0.02 |
| | C | 7.49 | 0.04 | - | - | - | 0.02 | - | - | - | 0.01 | 0.02 | 0.03 |
| | D | 6.78 | 0.14 | - | - | 0.02 | 0.06 | - | - | - | 0.02 | 0.07 | 0.11 |
| | E | 7.76 | 0.41 | 0.01 | - | 0.01 | 0.30 | 0.02 | 0.01 | 0.02 | 0.06 | 0.21 | 0.42 |
| | F | 8.12 | 0.57 | 0.04 | - | 0.06 | 0.33 | 0.07 | 0.01 | 0.02 | 0.06 | 0.23 | 0.59 |
| | G | 7.93 | 0.55 | 0.04 | - | 0.05 | 0.30 | 0.07 | 0.01 | 0.01 | 0.05 | 0.19 | 0.54 |
| | H | 7.63 | 0.09 | - | - | 0.01 | 0.05 | - | - | - | 0.02 | 0.04 | 0.07 |
| Glacial | I | 6.56 | 0.05 | - | - | 0.01 | 0.01 | - | - | - | 0.01 | 0.02 | 0.04 |
| lake | J | 6.83 | 0.07 | - | - | 0.03 | - | - | - | - | 0.01 | 0.02 | 0.04 |
| | K | 6.39 | 0.07 | - | - | 0.02 | - | - | - | - | 0.01 | 0.02 | 0.04 |
| | Min | 6.39 | 0.02 | - | - | - | - | - | - | - | 0.01 | 0.01 | 0.02 |
| | Max | 8.53 | 0.57 | 0.04 | | 0.06 | 0.33 | 0.07 | 0.01 | 0.02 | 0.06 | 0.23 | 0.59 |
| | Mean | 7.43 | 0.19 | 0.01 | - | 0.02 | 0.10 | 0.02 | - | 0.01 | 0.02 | 0.08 | 0.18 |
| | Std | 0.66 | 0.20 | 0.01 | - | 0.02 | 0.13 | 0.02 | - | - | 0.02 | 0.08 | 0.21 |
| | Median | 7.63 | 0.08 | - | - | 0.02 | 0.02 | - | - | - | 0.01 | 0.03 | 0.05 |
| | 25% quartile | 6.78 | 0.05 | - | - | 0.01 | 0.01 | - | - | - | 0.01 | 0.02 | 0.04 |
| | 75% quartile | 7.93 | 0.41 | 0.01 | - | 0.03 | 0.30 | 0.02 | 0.01 | 0.01 | 0.05 | 0.19 | 0.42 |

pH is dimensionless, EC units are in mS/cm and the rest in g/L, Std means standard deviation "-" means below the detection limit.

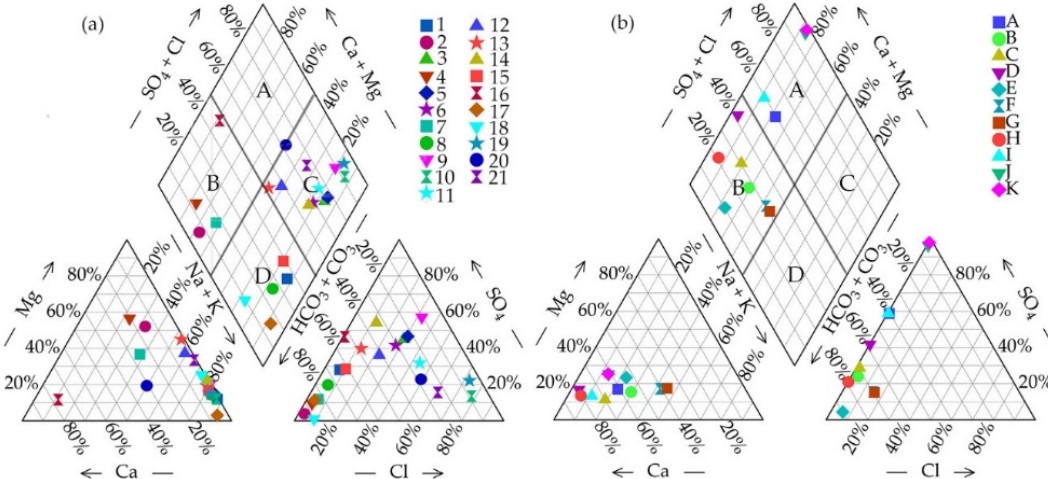

**Figure 2.** Piper diagram showing major ion concentration of the tectonic lakes (**a**) and glacial lakes (**b**).

Taken that the latitude of the tectonic lakes sorted from high to low as horizontal coordinates, and parameters and ion concentrations of the sample were vertical coordinates (Figure 3a), the tectonic lake data have two spatially distinct peak areas. The first peak appears in Xiao Qaidam Lake Region (19), which belongs to the desert arid and extremely arid climate of Qaidam Basin, with an average annual temperature of 1.1 °C, annual precipitation of 82.8 mm, and annual evaporation of 2031.8 mm, and the lake mainly relies on the Tatalin River, spring water, and groundwater supply. The second peak was located in the Dong Co (9) and Mubu Co (10), both belong to the Qiangtang alpine grassland semi-arid climate. The Dong Co was controlled by the Bangong Co-Nu Jiang River fracture zone, east and west of the lake for the marshes and saline land, south and north of the lake for the slope and flood of sand and gravel land. Overall, the lakes in both peak regions were closed inland lakes, where the water output only depends on evaporation, and the concentration makes the lake water physicochemical parameters and major ion concentrations higher, and in other regions, the physicochemical parameters and major ion concentrations were lower and moderate.

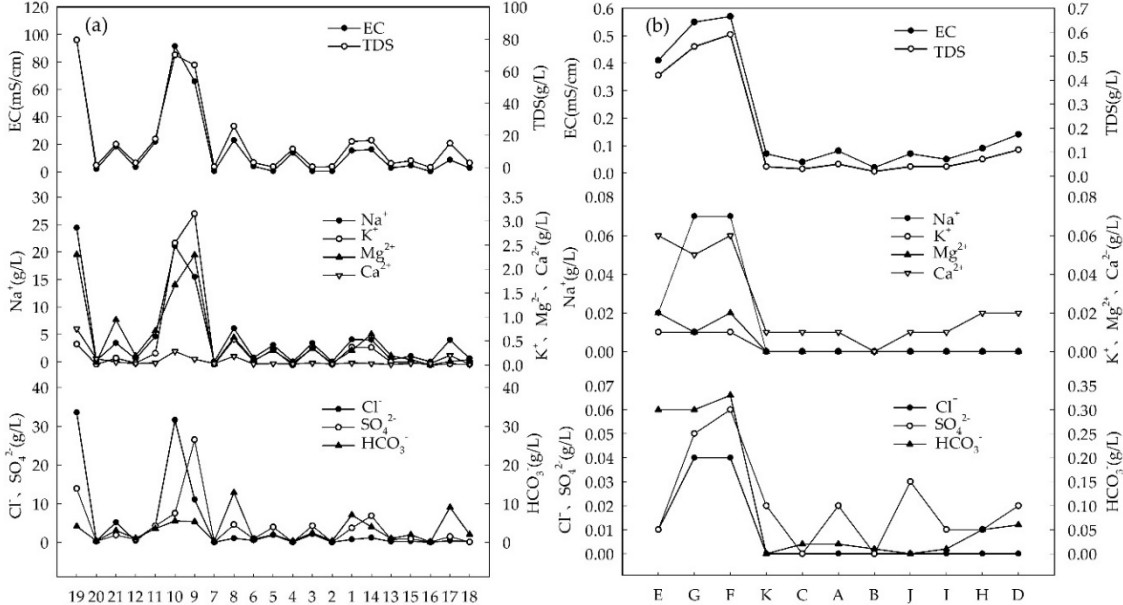

**Figure 3.** The distribution trends of TDS, Electrical Conductance (EC), and major ion concentration of Tectonic lakes (**a**) and glacial lakes (**b**).

### 4.2. Ion Characteristics of Glacial Lakes

The data of glacial lakes were much lower than that of tectonic lakes. The range of TDS were 0.02~0.59 g/L, with a mean value of 0.18 g/L. The ranges of pH, EC and TH were 6.39~8.53, 0.02~0.57 mS/cm (25 °C), and 0.01~0.23 g/L, respectively, and the mean value were 7.43, 0.19 mS/cm, and 0.08 g/L, respectively (Table 2). In Figure 2b, the cationic components of each samples were distributed in the lower left, indicating the predominance of $Ca^{2+}$ in cations. In the anions, the samples were mostly near the $HCO_3^-$ terminal, and followed by $SO_4^{2-}$. Moreover, 7 of the 11 glacial lakes were located in the Ca-HCO$_3$ (region B) and 4 in the Ca(Mg)-SO$_4$ (region A), indicating that the glacial lakes of the Tibetan Plateau are dominated by Ca-HCO$_3$ and Ca(Mg)-SO$_4$ types, and in the early and middle stages of lake evolution.

Latitude of the glacial lake samples from high to low as horizontal coordinates, and each data is in vertical coordinates (Figure 3b). We found that the physicochemical parameters and major ion concentrations were higher and more variable in the glacial lakes north of the K sampling point as a boundary, and south of K was decrease and level off significantly. The glacial lakes located in the Yairu Zangbo basin, its tributaries to the Geiqu basin developed a large area of modern glaciers (96, 131.32 km$^2$). The lakes of basin were mainly supplied by glacial meltwater and were open lakes [33], so the physicochemical parameters and major ion concentrations were low overall. The south of the K, the upper and middle reaches of the basin, with weak human activities and less tributary influx, and ions mainly come from rock weathering, hence the low ion content. The north of K belongs to the lower reaches of the basin, with increased human activities and tributaries flowing into the basin, resulting in higher ion content.

### 4.3. Sources of Hydrochemistry

#### 4.3.1. Ion Correlation

The ion concentration of lakes on the Tibetan Plateau varies greatly, and the correlation of ion concentration can reveal the similarity of hydrochemistry parameters in lakes as well as the consistency and differences in sources [34]. We calculated Pearson correlation coefficients for ion concentrations in tectonic lakes (Table 3) and glacial lakes (Table 4). In the tectonic lakes, $Cl^-$ was significantly positively correlated with $Na^+$, $Mg^{2+}$, and $Ca^{2+}$, with a correlation coefficient of 0.952 between $Cl^-$ and $Na^+$, and high correlation between $Cl^-$, $Na^+$ and TDS (r of 0.906, 0.977), indicating good commonality between $Cl^-$ and $Na^+$. The correlation coefficients between $SO_4^{2-}$ and $K^+$, $Mg^{2+}$ were 0.824 and 0.885 respectively, indicating that some of the ions in the lake water originated from sulfate dissolution, the poor correlation between $Ca^{2+}$ and $SO_4^{2-}$ indicates a general commonality of origin. The poor correlation between $NO_3^-$ and other ions indicated that there were differences in the origin of $NO_3^-$ from other ions, which may be derived from nitrification by soil microorganisms in addition to the $NO_3^-$ deposited in snow and ice [35]. In addition, there is an influence on the uptake of dissolved inorganic nitrogen by aquatic organisms [36], especially in high altitude lakes such as the Tibetan Plateau. $HCO_3^-$ was weakly positively correlated with other ions, which is related to the alkaline conditions of the tectonic lake, where $HCO_3^-$ may interact with cations to produce precipitation.

In the glacial lake, $HCO_3^-$ and $Mg^{2+}$, $Ca^{2+}$ significant positive correlation between the strongest, correlation coefficient of 0.994, 0.988, the low correlation coefficients between $Ca^{2+}$ and $SO_4^{2-}$ indicated poor co-origination. The coefficient between TDS and $Ca^{2+}$, $Mg^{2+}$, $HCO_3^-$ were also well, the above ions were the main components of the glacial lakes. This is due to the low TDS of glacial lakes (0.18 g/L) and the generally lower pH (7.43) than tectonic lakes (9.09), which makes sedimentation difficult.

**Table 3.** Pearson correlation coefficients for hydrochemistry parameters of tectonic lakes.

| | $Cl^-$ | $NO_3^-$ | $SO_4^{2-}$ | $HCO_3^-$ | $Na^+$ | $K^+$ | $Mg^{2+}$ | $Ca^{2+}$ | TDS |
|---|---|---|---|---|---|---|---|---|---|
| $Cl^-$ | | - | 0.570 ** | - | 0.952 ** | 0.588 ** | 0.847 ** | 0.837 ** | 0.906 ** |
| $NO_3^-$ | | | - | - | - | - | - | - | - |
| $SO_4^{2-}$ | | | | - | 0.758 ** | 0.824 ** | 0.885 ** | 0.470 * | 0.795 ** |
| $HCO_3^-$ | | | | | - | - | - | - | 0.455 * |
| $Na^+$ | | | | | | 0.714 ** | 0.933 ** | 0.833 ** | 0.977 ** |
| $K^+$ | | | | | | | 0.768 ** | - | 0.753 ** |
| $Mg^{2+}$ | | | | | | | | 0.700 ** | 0.944 ** |
| $Ca^{2+}$ | | | | | | | | | 0.791 ** |
| TDS | | | | | | | | | |

"**" indicates passing the 0.01 significant test, "*" indicates passing the 0.05 significant test, "-" indicates no significant correlation.

**Table 4.** Pearson correlation coefficients for hydrochemistry parameters of glacial lakes.

| | $Cl^-$ | $NO_3^-$ | $SO_4^{2-}$ | $HCO_3^-$ | $Na^+$ | $K^+$ | $Mg^{2+}$ | $Ca^{2+}$ | TDS |
|---|---|---|---|---|---|---|---|---|---|
| $Cl^-$ | | 0.860 ** | 0.892 ** | 0.998 ** | 0.833 ** | 0.884 ** | 0.854 ** | 0.941 ** | 0.941 ** |
| $NO_3^-$ | | | 0.647 * | 0.845 ** | 0.820 ** | 0.765 ** | 0.833 ** | 0.844 ** | 0.856 ** |
| $SO_4^{2-}$ | | | | 0.663 * | 0.848 ** | - | - | 0.671 * | 0.754 ** |
| $HCO_3^-$ | | | | | 0.911 ** | 0.957 ** | 0.994 ** | 0.988 ** | 0.991 ** |
| $Na^+$ | | | | | | 0.859 ** | 0.904 ** | 0.873 ** | 0.954 ** |
| $K^+$ | | | | | | | 0.966 ** | 0.927 ** | 0.941 ** |
| $Mg^{2+}$ | | | | | | | | 0.984 ** | 0.987 ** |
| $Ca^{2+}$ | | | | | | | | | 0.977 ** |
| TDS | | | | | | | | | |

"**" indicates passing the 0.01 significant test, "*" indicates passing the 0.05 significant test, "-" indicates no significant correlation.

### 4.3.2. Controlling Factors

Gibbs [37] mapped the relationship between TDS and $Na^+/(Na^+ + Ca^{2+})$ and TDS and $Cl^-/(Cl^- + HCO_3^-)$ in natural water bodies by analyzing the chemical composition of the world's rainwater, river, lake, and ocean water (Figure 4a), and pointed out that, under conditions weakly influenced by anthropogenic activities, the chemical composition of water bodies has three controlling factors: atmospheric precipitation, rock weathering, and evaporation–crystallization. In order to explore the major ion sources of lakes on the Tibetan Plateau, the detected data of the lakes in this study were plotted on the Gibbs diagram (Figure 4b). Tectonic lakes were far away from the atmospheric precipitation controlling region and highly concentrated at the evaporation–crystallization controlling region, except for the Daggyaima Co controlled by rock weathering. Glacial lakes concentrated in rock weathering control region (F and G controlled by evaporation–crystallization). These results indicated that tectonic and glacial lakes on the Tibetan Plateau are controlled by evaporation–crystallization and rock weathering, respectively.

### 4.3.3. The Ratio of Ions

The weathering of carbonate contributed more to the global river dissolution, accounting for 50%, 17.2% of evaporite rocks, and 11.6% of silicate [38]. It is believed that $Na^+$ and $K^+$ are mainly provided by evaporites or silicates, $Ca^{2+}$ and $Mg^{2+}$ may be from carbonates, evaporites as well as silicates, $HCO_3^-$ is mainly provided by carbonate rocks, and $Cl^-$ and $SO_4^{2-}$ are mainly from evaporites [39]. We can see that the samples of the tectonic lakes are mostly located on the lower side of the 1:1 line (Figure 5a), indicating that evaporite dissolution is the main source of tectonic lake ions; the samples of the glacial lakes are mostly located on the upper side of the 1:1 line, indicating that the ion source was mainly carbonate. The ratio of $(Ca^{2+} + Mg^{2+})/(Na^+ + K^+)$ in the water determines the weathering intensity of different rocks in the basin [17], which was influenced by carbonate weathering if the ratio was high, otherwise it

was controlled by evaporite. For example, the mean value of $(Ca^{2+} + Mg^{2+})/(Na^+ + K^+)$ in the world's river waters was 2.2, the Indus River Basin, which was mainly controlled by carbonate weathering, had a mean value of 6.0 [40], and the Taklimakan Desert River was dominated by evaporite dissolution (0.89) [41]. The tectonic lakes $(Ca^{2+} + Mg^{2+})/(Na^+ + K^+)$ ratios in this study ranged 0.07~6.20, with a mean value of 0.66 (Table 5). This indicated that the contribution of evaporite weathering to the ion composition of the tectonic lakes was outstanding, and the ratio of $(Ca^{2+} + Mg^{2+})/(Na^+ + K^+)$ was high (6.20) because of the outflow lake in Daggyaima Co, which showed the carbonate weathering control feature. The $(Ca^{2+} + Mg^{2+})/(Na^+ + K^+)$ ratio of the glacial lakes ranged between 0.88 to 16.50 with a mean value of 4.37, which was higher than the world river average (2.2) and lower than that of the Indus basin (6.0) [40]; thus, the ion composition was mainly influenced by the carbonate weathering of the basin.

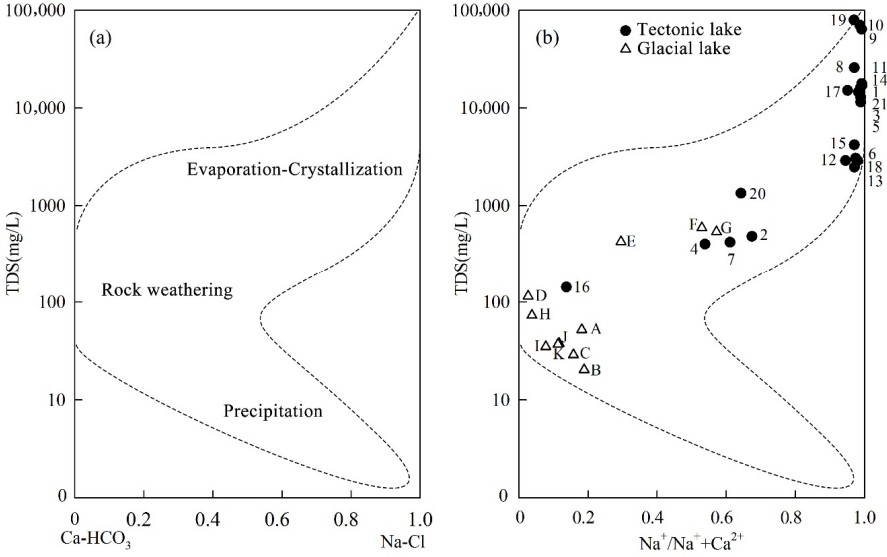

**Figure 4.** Gibbs distribution patterns (**a**) and the Gibbs diagram of tectonic and glacial lakes of the Tibetan Plateau (**b**).

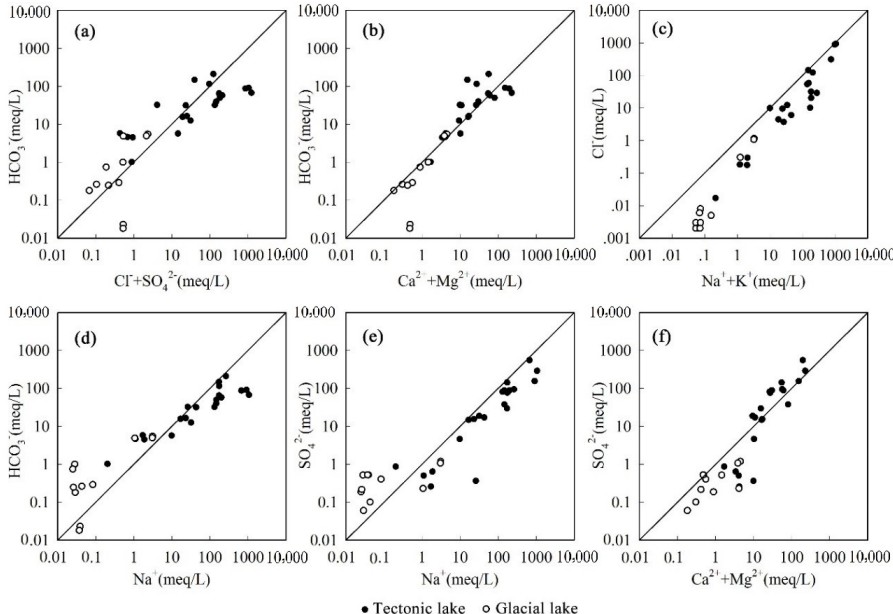

**Figure 5.** Major ion ratio of the tectonic lakes and the glacial lakes in Tibetan Plateau. The ratio of $HCO_3^-$ to $Cl^- + SO_4^{2-}$ (**a**), $HCO_3^-$ to $Ca^{2+} + Mg^{2+}$ ratio (**b**), $Cl^-$ to $Na^+ + K^+$ ratio (**c**), $HCO_3^-$ to $Na^+$ ratio (**d**), $SO_4^{2-}$ to $Na^+$ ratio (**e**), $SO_4^{2-}$ to $Ca^{2+} + Mg^{2+}$ ratio (**f**).

**Table 5.** The concentration ratio for $(Ca^{2+} + Mg^{2+})/(Na^+ + K^+)$ of the tectonic lakes and the glacial lakes in Tibetan Plateau.

| Type | Samples | $(Ca^{2+} + Mg^{2+})/(Na^+ + K^+)$ |
|---|---|---|
| Tectonic lake | Pung Co | 0.08 |
| | Bangkog Co | 1.14 |
| | Yaggain Co | 0.10 |
| | Co Ngoin | 1.96 |
| | Siling Co | 0.10 |
| | Serbug Co | 0.15 |
| | Qiagui Co | 1.04 |
| | Dagzê Co | 0.11 |
| | Dong Co | 0.13 |
| | Mubu Co | 0.08 |
| | Daxiong Lake | 0.15 |
| | Xueyuan Co | 0.38 |
| | Qido Co | 0.45 |
| | Dawa Co | 0.15 |
| | Daggyai Co | 0.13 |
| | Daggyaima Co | 6.20 |
| | Ngamring Co | 0.07 |
| | Lang Co | 0.20 |
| | Xiao Qadam Lake | 0.12 |
| | Gahai Lake | 0.74 |
| | Qinghai Lake | 0.28 |
| | Mean | 0.66 |
| Glacial lake | A | 2.10 |
| | B | 1.36 |
| | C | 2.82 |
| | D | 16.50 |
| | E | 2.33 |
| | F | 1.02 |
| | G | 0.88 |
| | H | 8.99 |
| | I | 4.58 |
| | J | 3.73 |
| | K | 3.80 |
| | Mean | 4.37 |

In Figure 5b, all samples were close to 1:1 line, indicating that $Ca^{2+}$, $Mg^{2+}$ and $HCO_3^-$ in tectonic lakes and glacial lakes mainly originate from the dissolution of calcite ($CaCO_3$) and dolomite ($CaMg(CO_3)_2$). Individual glacial lake samples were located on the lower side of 1:1 line, indicating that silicate dissolution contributes to $Ca^{2+}$ and $Mg^{2+}$ in glacial lakes. When the $Na^+ + K^+$ to $Cl^-$ ratio of water is 1:1, it means that ion composition is controlled by the dissolution of evaporite [1]. The relationship between $Na^+ + K^+$ and $Cl^-$ showed that the samples of tectonic lakes were close to the 1:1 line (Figure 5c), indicating that the rock salt (NaCl) and potassium salt (KCl) are the main sources of $Na^+$, $K^+$ and $Cl^-$ for tectonic lake. The glacial lake samples were all below the 1:1 line, implying that $Cl^-$ is not sufficient to balance $Na^+$ and $K^+$, and remaining $Na^+$ and $K^+$ may originate from the dissolution of silicates (potassium, sodium feldspar, etc.). Figure 5d,e illustrates that the dissolution of sulfate minerals may also be another important source of $Na^+$ in tectonic lakes. Compared with Figure 5e, the glacial lake samples were closer to 1:1 line of $Ca^{2+}+Mg^{2+}$ and $SO_4^{2-}$ (Figure 5f), indicating that the dissolution of sulfate minerals is one of the sources of $Ca^{2+}$, $Mg^{2+}$ and $SO_4^{2-}$ in glacial lakes.

## 5. Conclusions

The tectonic lakes of the Tibetan Plateau had much higher physicochemical parameters and ion concentrations than the glacial lakes, and TDS fluctuated from high to low latitudes. The variations of

ion content in the northern part of the Qiagui Co were more fluctuating and had two obvious peaks, while the variations in the southern part were moderate. The TDS of glacial lakes were low, and leveled off in the upper and middle reaches of the basin, while lower reaches were higher and more variable.

The tectonic lakes of the Tibetan Plateau were mainly chloride saline lakes, with $Na^+$ as major cation, and $SO_4^{2-}$, $Cl^-$ as major anions. Glacial lakes were mainly carbonate and sulfate type lakes, $Ca^{2+}$ and $Mg^{2+}$ were the major cations, the $HCO_3^-$ was the major anion, followed by $SO_4^{2-}$.

The hydrochemistry process of tectonic lakes were mainly controlled by evaporation–crystallization, and the ions mainly came from the evaporite of the basin, in which NaCl and KCl were the main sources of $Na^+ + K^+$ and $Cl^-$ in the lake water. Glacial lakes were mainly influenced by the weathering of basin rocks, with ion sources strongly influenced by the weathering of carbonates than evaporites, with calcite and dolomite being important sources of $Ca^{2+}$, $Mg^{2+}$ and $HCO_3^-$.

**Author Contributions:** Conceptualization, M.S.; methodology, M.S., H.J., L.Y.; software, H.J., L.Y.; formal analysis, X.Y.; investigation, X.L., Y.G.; writing—original draft preparation, H.J., L.Y.; writing—review and editing, M.S., X.Y.; visualization, H.J., L.Y.; supervision, M.S., X.Y.; project administration, M.S.; funding acquisition, M.S., X.Y. All authors have read and agreed to the published version of the manuscript.

**Funding:** National Natural Science Foundation of China (41861013, 41801052), and Chinese Academy of Sciences "Light of West China" Program.

**Acknowledgments:** We would like to thank our colleagues at Northwestern Normal University for their help with the writing process. We would like to thank those who helped us with the experiment. We thank the anonymous reviewers and editors for their constructive and helpful suggestions.

**Conflicts of Interest:** The authors declare no conflict of interest.

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
