# Peer review of "Hydrochemistry Differences and Causes of Tectonic Lakes and Glacial Lakes in Tibetan Plateau"

_water, doi:10.3390/w12113165_

Round 1
Reviewer 1 Report
The paper characterizes the hydrochemical conditions of 32 lakes located in the Tibetan Plateau. Their exact localization in the figure presented is very difficult. GPS coordinates should be provided. The choice of study area is interesting as it is a poorly recognized limnologically area. I think that methods used in the study are appropriate. Also laboratory analyzes are not questionable. It is a pity that the research was limited to one-time sampling. On the basis of one-off measurement, we cannot conclude about the seasonal variability of physicochemical parameters of waters. In is frequentely stated in other studies, that the seasonal variability can be very high. The authors divided the lakes into two groups of tectonic and glacial lakes. In many cases, the lake's genesis influences the hydrochemical conditions. However, in this case, the key role is played by the climatic conditions in the lake's catchment area (in particular the presence of glaciers or long-term snow patches, which are the source of low-mineralized waters). It is most likely visible in the case of the Daggyai tectonic lake with an EC of 0.18 mS / cm-1. However, in the absence of GPS coordinates, I am not able to say whether it is powered by a melting glacier. The interpretation of the results and the conclusions drawn are correct. These conclusions are quite obvious - glacial lakes fed from melting glaciers and snow will always have a low mineralization of water and the dominance of HCO3- and Ca2 + ions. In turn, in lakes of semi-deserts and deserts, Cl- and Na + ions will dominate. The authors also drew such conclusions.
Despite the one-time sampling (which is the biggest drawback of this work) and quite obvious conclusions, I believe that the article deserves to be published. It brings new information about the hydrochemistry of lakes, poorly known in terms of limnology, of the Tibetan Plateau.
I have only minor issues to be addressed.
Why are there differences in number of samples per a lake (2 - 6)? Due to size of lake?
Was normality of distribution checked prior Pearson correlation test?
In general, I miss subchapter of material and methods devoting to statistical analyzes and visualization (correlation test, Piper diagram, Gibbs diagram etc).
If you provide arithmetic means give also standard deviation
Table 4, in the sequence in the text, has no caption.
Line 154 write pH instead of PH
Author Response
Dear reviewers,
Thank you very much for your approval of our manuscript and your valuable suggestions. According to your review comments, we have revised the manuscript, the modification is described below.
First, in response to your question about the lack of GPS coordinates in the manuscript, we have added Table 1 to the manuscript, showing the GPS coordinates for each sampling location and the relevant parameters for constructing the lake.
The following are responses to a few minor issues you have raised:
- We collected 2-6 samples for each lake, depending on the size of the lake. 2 samples were collected with an area of <100 km2, lake area between 100~500 km2 with 4 samples, area larger than 500 km2 with 6 samples. Finally, taken the average of the samples data after testing.
- We have checked the normality of the ion distribution before testing the pearson correlation. Tectonic lakes showed a positive bias, with a bias of 1.6 to 3.3, and peaks mostly showed thick peaks (>3). Glacial lakes also showed a positive bias, with a bias of about 1, and peaks close to 0.
- According to your opinions, we have added 3.2 piper diagram, 3.3 ion correlation analysis, 3.4 gisbbs diagram, 3.5 ion ratio in the third part of the article (materials and method). Explained statistical analysis methods and the process of creating graphs. Please see the revised draft for details of revisions.
- We have added the minimum, maximum, median, standard deviation, upper and lower (25%, 75%) quartiles of each indicator to Table 2 (original Table 1) in order to better characterize the numerical changes.
- We have added the caption to Table 5 (original Table 4), “The concentration ratio for (Ca2++Mg2+)/(Na++K+) of the tectonic lakes and the glacial lakes in Tibetan Plateau” in row 294-295.
- We have corrected the writing errors in the original manuscript regarding pH. Row 130, 202 (original row 159).
Thank you again for your valuable comments on this article, and we will continue to improve it if you have any other questions.
Thanks for all the help.
Best wishes.
Reviewer 2 Report
Dear authors
This is an interesting study, as this type of hydrologic and chemical environment is not discussed very often in the scientific literature. Furthermore, I am impressed by the amount of time and travel invested by the authors of the paper in their effort to collect samples for water chemistry testing.
A review of the hydrologic and chemical calculations provided in the paper shows that they are usually correct from a mathematical point of view. Nevertheless, from a hydrochemistry point of view, it would be worthwhile to examine the way HCO3 concentration calculations are performed in the paper.
Short commentary and a question are provided below:
Comment no. 1 concerns the method of calculating bicarbonate (HCO3) given in the paper. The HCO3 concentration calculation method provided by the authors, as the difference between the cation total and anion total, is frequently used and it is a generally correct and adequate method for so-called regular water chemistry, defined as water with a pH of 6.2 and 8.3. However, the water described in the submitted paper has a pH of around 10, which means that the concentration of CO3 cannot be neglected. It is important to note that at a pH of 10.4 the concentration of CO3 equals that of HCO3.
Question:
Given the high mineral content of the studied water, why did the authors not perform a simple titration of the samples? This would have allowed them to determine the mineral alkalinity and total alkalinity of the studied samples. In turn, this would have made it easy to calculate the HCO3 concentration with a high degree of accuracy. Finally, it would have been possible to balance the chemistry results and calculate the real error for water chemistry.
Comment no. 2: Editing errors need to be fixed – subscripts and superscripts of ions listed in the paper; for example, SO42+_ should be written as SO42- (and so on). Editing errors are marked in the paper using colour. The caption for the table is missing inline 245. The line contains some text accidentally copied from somewhere else.
It is better to list electrolytic conductivity (EC) values using micro-Siemens per centimetre. The water temperature at which EC was measured should also be provided in the methods section, e.g. EC25oC. This is an important piece of information, as EC values strongly depend on water temperature.
Comment no. 3 concerning Table no. 1.
a. The number of significant digits provided by the authors is very large, which is difficult to accept given the 5% error listed in the paper. Please do not list every ion concentration using notation with two places after the decimal point. Please fix Table no. 1 and propose a more practical notation system for it. For example, it is difficult to accept the Cl concentration for Daggyaima Co = 0.59 mg/L, in light of the CL concentration provided for Mubu Co: 31558.74 mg/L.
b. In cases where the concentration of a given ion is not given (e.g. NO3), inform the reader of the paper that the concentration of the ion was below the limit of detection (e.g. <L.D.).
Comment no. 4. The calculation of an average value is appropriate only in cases where we are certain that the empirical data set has a normal distribution. In Table 1 the distribution of concentrations of each ion is asymmetric, in which case I would recommend calculating a few other measures, such as the median and upper and lower quartiles for each given ion concentration, in addition to the average value. The three “additional” measures mentioned above are very useful in describing hydrologic and chemical environments characterized by a large range of variability.
Comment no. 5 concerning the calculation of the correlation coefficient (section 4.3.1).
a. The authors most likely computed the coefficient of correlation “r” basing on ion concentrations given in Table 1. Pearson’s correlation coefficient “r” describes linear relationships, which prompts the following question: Did the authors check the compatibility of the their empirical data (i.e. ion concentrations) with the normal distribution? If the ion concentration distribution is not consistent with a normal distribution, then the authors should explain how they prepared their data for further calculations. Pearson’s correlation coefficient is sensitive to outliers.
b. In tables no. 2 and 3 the authors provide “r” correlation values for the significance level p = 0.05 and p = 0.01. Two comments are relevant here. First, in Table 1 please leave only significant correlations. I suggest that the authors insert a special symbol into Table 1 (symbol: “–“). The symbol should be explained in a footnote in the following way: “symbol means no significant correlation.” Second, it is only necessary to discuss data at the p = 0.05 significance level. Finally, please do not insert the value “1” diagonally across tables given in the paper.
In chapter no. 3 (Materials and Method) please add a table listing sites and their ID from the map as well as geologic structure, elevation above sea level, lake parameters, information on the climate zone, information on the studied elevation zone, and perhaps some meteorologic information, such as average air temperature. If these types of information are provided in the paper, then this study would be useful in comparisons with other climate zones with a known local geologic structure. Of course, it is also possible to describe some other additional characteristics of the study area, such as local hydrogeology.

Author Response
Dear reviewers,
Thank you very much for your approval of our manuscript and your valuable suggestions. According to your review comments, we have revised the manuscript, the modification is described below.
- Obviously, this is a very reasonable suggestion. For higher pH waters, the CO32- concentration cannot be ignored and the most logical way to determine it is to use a titration method. It is a pity that we did not use the titration method for sample detection and we used a Dionex-300 ion chromatograph for the anion detection.Since the eluent used for the anion separation column was a Na2CO3/NaHCO3 solution, it could not be used for the CO32- determination. We extrapolated the HCO3- content using conservation of charge, and since the vast majority of tectonic lakes have a pH < 10.3, the HCO3- content is greater than the CO32- content, and since the HCO3- content estimated by conservation of charge is greater than the actual value, this may contain CO32-. There may still be some deficiencies in our work and we would like to be corrected by the reviewers.
- We have checked the manuscript and corrected some edit errors in the text, such as row 25, 26 (original row 21, 22), row 135 (original row 118), row 140 (original row 123), row 172-173 (original row 131), row 178 (original row 136), row 200 (original row 118), row 203 (original row 160), row 223 (original row 179), row 237 (original 192), row 320, 324 (original row 269,273), and Figure 3 (row 227-229). We have corrected the edit errors in the original manuscript regarding pH, in row 130, 202 (original row 159). We have added the caption to Table 5 (original row 245), “The concentration ratio for (Ca2++Mg2+)/(Na++K+) of the tectonic lakes and the glacial lakes in Tibetan Plateau” in row 294-295.
We have noted the temperature on the detection of EC at 25 ° C, and added the temperature information for EC detection (row 197). In addition, we have added detailed information in the manuscript for sample collection, storage, and detection (row 186-190, row 195-201).
- We have revisied the ion concentration units in Table 2 (original Table 1) to g/L and annotated the non-displayed ion concentrations as "-", indicating that ion was below the detection limit (row 221-224).
- In your opinion, we have added the minimum, maximum, median, standard deviation, upper and lower (25%, 75%) quartiles of each indicator to Table 2 (original Table 1) in order to better characterize the numerical changes.
- a. We have checked the normality of the ion distribution before testing the pearson correlation. Tectonic lakes showed a positive bias, with a bias of 1.6 to 3.3, and peaks mostly showed thick peaks (>3). Glacial lakes also showed a positive bias, with a bias of about 1, and peaks close to 0. Based on this, we conducted the Pearson correlation coefficient test.
b. We have removed the diagonal "1" values from the Table3, 4 (original Table 2, 3) and denoted no significant correlation with "-", retaining significant correlations and and describing them.
Finally, with reference to your comments, we have added Table 1 to the Study area section (row 105-110), adding relevant parameters such as the location of the mass center of the sampled lake, the geological region to which the lake belongs, elevtion, lake area, recharge coefficient, average annual precipitation, and average annual temperature. The relevant parameters are taken from The Lake of China [25].
Thank you again for your valuable comments on this article, and we will continue to improve it if you have any other questions.
Thanks for all the help.
Best wishes.
Reviewer 3 Report
MDPI
Manuscript Number: water-963441
Title: Hydrochemistry Differences and Causes of Tectonic Lakes and Glacial Lakes in Tibetan Plateau
Article Type: Research Article
Keywords: hydrochemistry process; major ion characteristics; tectonic lakes; glacial lakes; Tibetan 29 Plateau
_______________________________________________________________________________
GENERAL NOTES
The manuscript aims to investigate the differences in hydrochemistry between tectonic lakes and glacial lakes on the Tibetan Plateau, the largest lake cluster in China and the world. To do that, the Authors have sampled 32 lakes in an area where the number of lakes is several hundred (cfr. Wan et al. 2026. A lake data set for the Tibetan Plateau from the 1960s, 2005, and 2014. Nature, DOI: 10.1038/sdata.2016.39). It is therefore obvious to ask how significant those 32 lakes are compared to the entire population of lakes. In particular if we consider that the paper does not go into detail on how the lakes were chosen. In other words: how are representative the studied lakes? The Authors must put particular attention on this question and explain well the monitoring plan.
COMMENTS
The first comment is on English language that is poor. There are parts of the manuscript that are difficult to understand. In particular the sentences corresponding to these rows require a language revision: 19-21; 36-38; 81-85; 150-156; 170-176; 200-204; 265-267.
Regarding the content or the methodology it is important to underline:
- the sampling procedure is not well described, in addition to what was said previously on representativeness: i.e. a) where are sampled the lakes: on the shore line? b) which kinds of samples conservations was adopted in the transport to the laboratories? c) how long was the conservation before the analysis? The last is particular important for the nitrate measurements, etc.;
- it is not clear what means “to the size of the lake” at row 106;
- the reference temperature for conductivity measurements must always be indicated (for example: 20 ° C. Conductivity varies of 2% for each degree centigrade). See rows: 131, 160 etc.;
- put attention on the units: ms/cm is wrong. Please change in mS/cm (S = Simens!!!)
- in a couple of cases the Authors reported “… stage of lake evolution”. It is not clear which kind of evolution they consider: trophic, geochemical etc. In any case it is requested a clear definition;
- In the discussion of the poor correlation between NO3 and others ions please consider the biological role un the aquatic biocenosis. In high altitude lakes, like those of Tibetan Plateau, the very low levels of nitrate concentrations are related to a very low atmospheric load and to the assimilation by the algae in the lakes;
- please clarify the sentence “This is due to the low TDS of glacial lakes (177.24 mg/L) and the generally lower pH (7.43) than tectonic lakes (9.09), which makes sedimentation difficult.” (Rows 203-204) What role has the pH?
DETAILED INDICATIONS
It is suggested to introduce these modifications in the manuscript:
- rows 86-101: use a bulleted list;
- row 103: please explain what means G315, S301; …
- row 118-119: please describe better what means “The accuracy of ion determination was ng/g”. ng/g or ng/L? What about the Limit of detections, precisions etc. Just some indications to delineate the analytical quality procedures;
- row 160: change PH in pH;
- row 177: it is necessary to put attention on the layout of the table
- Table 1: it is suggested to:
- indicate if the nitrate, sulfate and hydrogenocarbonate are: mg N/L or mg NO3/L; mg S/L or mg SO4/L, meq/L or …, respectively;
- use 2 decimals for pH, including 0 (i.e. 9.9 change in 9.90 etc.)
- reduce the number of decimals according to the amount of concentrations. It is possible it is suggested to change the units, where appropriate, in g/l or express all concentrations in meq/L.
- Figure 4a: in the x axe the label is probably Cl/(Cl+HCO3). It is correct?
- row 245: please insert the caption!
Author Response
Dear reviewers,
Thank you very much for your approval of our manuscript and your valuable suggestions. According to your review comments, we have revised the manuscript, the modification is described below.
For the selection of lakes, we took into account the geological structure of the Tibetan Plateau at the beginning of the experimental design. The Tibetan Plateau has nine stratigraphic zones in the Tibet Autonomous Region and Qinghai Province, and we selected and sampled 21 tectonic lakes and multiple glacial lakes in four stratigraphic zones along the route, taking into account the accessibility of the route and lake locations. The sampling of glacial lakes in particular was a very arduous task, even requiring reaching altitudes above 5000m. The lakes we sampled may not be a good representation of the entire Tibetan Plateau lake group, but the differences between the cognitive tectonic lakes and glacial lakes should be of some significance. We hope that the reviewers will understand and sincerely thank you for your understanding.
- We have made some revisions to the language issues in the text, especially with regard to your pointed: row 23-25; 40-42; 85-88; 193-199; 231-220; 247-251; 316-318 (original row19-21; 36-38; 81-85; 150-156; 170-176; 200-204; 265-267). Make it possible to express the our intention more clearly.
- We have noticed this problem, and we have added detailed information in the manuscript for sample collection, storage, and detection (row 186-190, row 195-201).
- In the manuscript of row 117 (original row 106) means the number of lake samples is based on the area of lake. We collected 2-6 samples for each lake, 2 samples were collected with an area of <100 km2, lake area between 100~500 km2 with 4 samples, area larger than 500 km2 with 6 samples. Finally, taken the average of the samples data after testing.
- We have noted the temperature on the detection of EC at 25 ° C, and added the temperature information for EC detection (row 197). For the edit errors in EC units, we have revised at row 173, 203, 223 (original 131, 160, 179), and Figure 3 respectively.
- The evolution of the Tibetan Plateau lakes, generally experienced from freshwater lakes to saline lakes and then to salt lakes, until the end of the dry salt lake evolution, along with its hydrochemistry type also changed, range from carbonate-sulphate-chloride types. In this paper, we measure the evolution stage of the lake according to the main ion types in the lake. The revisions were shown in row 269-371.
- We considered the impact of NO3 uptake by aquatic organisms, which has been revised as “in addition, there is an influence on the uptake of dissolved inorganic nitrogen by aquatic organisms [36], especially in high altitude lakes such as the Tibetan Plateau”.
[36]. Gibson, C.A; O’Reilly, C.M; Conine, A.L.; Lipshutz, S.M. Nutrient uptake dynamics across a gradient of nutrient concentrations and ratios at the landscape scale. Journal of Geophysical Research: Biogeosciences 2015, 120(2), pp. 326-340. doi: 10.1002/2014JG002747.
- At row 205-251 (original row 203-204), we would like to express that HCO3- may interact with cations to form precipitates when under alkaline conditions. Since the pH of the glacial lake is neutral, it is difficult to convert to CO32- to form precipitation.
Detailed revisions:
- We have added Table 1 to the manuscript, indicating the stratigraphic regions in which each lake is located, and describing the associated stratigraphic region at row 89-101. We also added the coordinates of the lake's center of mass, lake area, and other relevant parameters.
- The “G” of “G315” indicates a national highway and the “S” of “S301” indicates a provincial road, which we have revised in the manuscript accordingly. For details row113-114.
- The expression in the manuscript may be inappropriate, as ng/g refers to the accuracy of ion detection by ion chromatography and indicates how many ng of solutes can be detected per g of solution. We have revised the structure of the sentences.
- We have revised the edit error in the manuscript regarding pH, at row 130, 202 (original row 159).
- We have made appropriate revisions to the table structure in the manuscript, as shown in Table 2 (original Table 1), Table 3 (original Table 2), Table 4 (original Table 3), Table 5 (original Table 4). In Table 2, the width of the table was taken into account, as shown in the table, and the units were indicated below the table.
We have used two decimals for pH and complemented it with “0”.
We have changed the unit of ion concentration in Table 2 to g/L and made corresponding changes in the manuscript. Gibbs diagram (Figure 4) to not affect the analysis of the results, the unit of ion concentration remains mg/L.
- Obviously, your advice feich reasonable. Gibbs mapped the relationship between TDS and Na+/(Na++Ca2+) and TDS and Cl-/(Cl-+HCO3-) in natural water bodies. We consider that HCO3 was derived from conservation of charge and not an actual detection value, which was not a good representation of the ion controlling factors, so only the relationship between TDS and Na+/(Na++Ca2+) was shown in Figure 4.
- We have added the caption to Table 5 (original row 245), “The concentration ratio for (Ca2++Mg2+)/(Na++K+) of the tectonic lakes and the glacial lakes in Tibetan Plateau” in row 294-295.
Thank you again for your valuable comments on this article, and we will continue to improve it if you have any other questions.
Thanks for all the help.
Best wishes.
Round 2
Reviewer 2 Report
Dear Authors,
Thank you for making an effort to improve the manuscript. Most of your responses are quite thorough and completely acceptable. There is one issue that I would like you to still clarify. The problem area is noted in Comment no. 5a (cited below).
Comment no. 5 concerning the calculation of the correlation coefficient (section 4.3.1). The authors most likely computed the coefficient of correlation “r” basing on ion concentrations given in Table 1. Pearson’s correlation coefficient “r” describes linear relationships, which prompts the following question: Did the authors check the compatibility of the their empirical data (i.e. ion concentrations) with the normal distribution? If the ion concentration distribution is not consistent with a normal distribution, then the authors should explain how they prepared their data for further calculations. Pearson’s correlation coefficient is sensitive to outliers.
I need to check your calculations; therefore, please send data sets for Tables 3 and 4: Ca-SO4; TDS–HCO3; Mg–SO4.
In addition, please provide the elevation above sea level and calculate the relationship between TDS and elevation above sea level.
It would also be worthwhile to take another look at the conclusion provided in the first sentence (in bold):
"The tectonic lakes of the Tibetan Plateau had much higher physicochemical parameters and ion concentrations than the glacial lakes, and TDS was decreasing from high to low latitudes."
This type of conclusion can be proven using a coefficient of correlation. Otherwise, it is only speculation.
Truly Yours
Author Response
Dear reviewers,
Thank you for taking the time to review the changes we have made to this article. With regard to your question, we previously checked the normality of the ion concentration distribution. As previously replied, the tectonic lakes showed a positive bias, and peaks mostly showed thick peaks (>3). Glacial lakes also showed a positive bias, with a bias of about 1, and peaks close to 0. Based on this, we conducted the Pearson correlation coefficient test.
In the original manuscript, our description was that we thought the tectonic lake showed a decreasing trend from high to low latitude TDS, and after our correlation analysis of TDS, latitude, and elevation, we found no significant correlation between them.
We found that our previous description was inaccurate, and the TDS of the tectonic lake can only be described as fluctuating from high latitude to low latitude. Therefore, we have changed the manuscript accordingly. Row 21, 314.
In the meantime, we have provided the data from Tables 3 and 4 (in the Appendix), which provide the elevation and latitude of the relevant locations.
Thank you again for your valuable comments on this article, and we will continue to improve it if you have any other questions.
Thanks for all the help.
Best wishes.